# Polydopamine-Lysophosphatidate-Functionalised Titanium: A Novel Hybrid Surface Finish for Bone Regenerative Applications

**DOI:** 10.3390/molecules25071583

**Published:** 2020-03-30

**Authors:** Fiona Baldwin, Tim J. Craig, Anna I. Shiel, Timothy Cox, Kyueui Lee, Jason P. Mansell

**Affiliations:** 1Department of Applied Sciences, University of the West of England, Bristol BS16 1QY, UK; fiona2.baldwin@live.uwe.ac.uk (F.B.); tim.craig@uwe.ac.uk (T.J.C.); anna.shiel@uwe.ac.uk (A.I.S.); timothy.cox@uwe.ac.uk (T.C.); 2Department of Bioengineering, University of California, Berkeley, CA 94720, USA; kyueui@berkeley.edu

**Keywords:** polydopamine, lysophosphatidic acid, titanium, functionalisation, osteoblasts, differentiation

## Abstract

Aseptic loosening of total joint replacements (TJRs) continues to be the main cause of implant failures. The socioeconomic impact of surgical revisions is hugely significant; in the United Kingdom alone, it is estimated that £135m is spent annually on revision arthroplasties. Enhancing the longevity of titanium implants will help reduce the incidence and overall cost of failed devices. In realising the development of a superior titanium (Ti) technology, we took inspiration from the growing interest in reactive polydopamine thin films for biomaterial surface functionalisations. Adopting a “one-pot” approach, we exposed medical-grade titanium to a mildly alkaline solution of dopamine hydrochloride (DHC) supplemented with (3S)1-fluoro-3-hydroxy-4-(oleoyloxy)butyl-1-phosphonate (FHBP), a phosphatase-resistant analogue of lysophosphatidic acid (LPA). Importantly, LPA and selected LPA analogues like FHBP synergistically cooperate with calcitriol to promote human osteoblast formation and maturation. Herein, we provide evidence that simply immersing Ti in aqueous solutions of DHC-FHBP afforded a surface that was superior to FHBP-Ti at enhancing osteoblast maturation. The facile step we have taken to modify Ti and the biological performance of the final surface finish are appealing properties that may attract the attention of implant manufacturers in the future.

## 1. Introduction

Titanium (Ti) is a widely used bone implant material. The popularity of Ti stems from its excellent biocompatibility, corrosion resistance and high modulus of elasticity in tension [1]. Total joint prostheses for replacing diseased hips and knees are fashioned from Ti. In the United Kingdom, a total of 106,111 hip and 109,540 knee primary arthroplasties were performed in 2019 (Prof. AW Blom, data analysis licensee for the National Joint Registry). Of these, it is estimated that 7%–8% will need replacing due to aseptic loosening, a major cause of joint replacement failure [2], generally thought to be a consequence of poor initial and/or sustained osseointegration. The impact of aseptic loosening is huge, with approximately 55% of hip [3] and 31% of knee revisions [4] attributed to the problem. Consequently, there has been a wealth of research interest in seeking ways of improving osseointegration. Such approaches include the development of Ti technologies that promote more rapid and superior rates of integration by enhancing, for example, the activity of bone-forming osteoblast cells at the Ti surface [5]. 

In recent years, we have focussed on a stable lysophosphatidic acid (LPA) analogue as a potential adjunct for enhancing early osseointegration of bone implant materials, including Ti [6,7]. The agent in question is (3S)1-fluoro-3-hydroxy-4-(oleoyloxy)butyl-1-phosphonate (FHBP), which we discovered synergistically co-operated with calcitriol (1,25D) to bolster human osteoblast maturation [8]. In a biomaterials context, this bioactive lipid is particularly attractive given its small size and stability; in contrast to much larger, bulkier, protein growth factors, e.g., BMP-2, we find FHBP-coated Ti could withstand autoclaving [7] and 35kGy gamma irradiation [9]. It is also likely, given its structural similarity to related lipids (e.g., oleic acid), that FHBP will withstand prolonged ambient storage (potentially 2–3 years), as would be expected for routinely stored implantable devices. Credence is given to this possibility in light of our recent study, wherein FHBP-coated Ti retained the ability to enhance 1,25D-induced osteoblast maturation after 24 months of ambient storage in air [9].

As part of our ongoing programme to bio-functionalise bone biomaterials with LPA species, we have recently turned our attention to polydopamine (PDA), a versatile reactive platform that is gaining significant momentum in (bio)material design [10,11,12]. Inspired by the tenacity with which the edible blue mussel (*Mytilis edulis)* attaches to wet surfaces, Messersmith’s team [13] reported on the facile and robust ability of PDA thin films to deposit at virtually any surface, including Ti. The PDA assembly is triggered by oxidative oligomerisation of dopamine, producing indole derivatives that can be further cross-linked or physically stacked by noncovalent interactions [14,15]. The multifunctionality of PDA relies on the unique catechol (1,2-dihydroxybenzene) moiety, which not only can deposit metal layers by reducing metallic ions [16] but also allows secondary surface modification via various chemical interactions [12]. It has since emerged that the interaction of deposited PDA films with Ti is one of the strongest noncovalent interactions to date; using atomic force microscopy, Lee et al. [17] found that the catechol moieties of PDA bound to wet TiO_2_ with a dissociation energy of 22.2 kcal mol^-1^. The step taken to functionalise Ti with PDA is unremarkable; simply steep specimens in mildly alkaline (~ pH8–9) solutions of dopamine hydrochloride (DHC) and, within a matter of minutes to hours, robustly bound reactive films form at the surface [12]. Resultant PDA films can then be tailored to carry a plethora of biocompatible, bioactive species to suit the particular application. Alternatively, (bio)material surface functionalisations can be achieved using a “one-pot” approach by combining DHC with the selected agent of interest and then applying the resultant solution(s) to the test surface.

Thus far, the direct conjugation of LPA/LPA analogues to PDA has not been forthcoming, but there are reports that PDA platforms can support the attachment/integration of lipids; liposome-PDA surface coatings have been fabricated to explore the delivery of fluorescent cargo to myoblast cells [18]. Using mica substrates, Nirasay et al. [19] used PDA to support a phospholipid bilayer composed of dimyristoylphosphatidylcholine and dioleoylphosphatidylcholine. Most recently, Ding and co-workers developed helically coiled and linear PDA nanofibers using lipid nanotubes as templates [20]. In their “hot paper”, Ding et al. employed oleic acid to enhance the association of the selected lipid with PDA [20]. 

Herein, we report on the facile development of a bioactive Ti substrate whereby PDA reactive films capture FHBP. Adopting a “one-pot” approach, we initially utilised a tissue culture plastic (TCP) model to help inform further direction towards Ti surface functionalisation, and some of these findings are also presented.

## 2. Materials and Methods

### 2.1. General 

Unless stated otherwise, all reagents were of analytical grade from Sigma (Poole, UK). Stocks of 1,25D (100 μM) were prepared in ethanol and stored at −20 °C. The lysophosphatidic acid (LPA) analogue, (3S)1-fluoro-3-hydroxy-4-(oleoyloxy)butyl-1-phosphonate (FHBP), was obtained from Tebu-Bio (Peterborough, UK) and reconstituted to 500 μM in 1:1 ethanol:cell culture-grade water and stored at −20 °C. A bicinchoninic acid (BCA)-based assay kit (Pierce) was from Fisher Scientific UK Ltd. (Loughborough, UK), and the detecting reagent mixture was prepared exactly as instructed on the day of use. Grit-blasted and acid-etched medical-grade titanium discs (10-mm diameter) were kindly provided by OsteoCare (Slough, UK).

### 2.2. Maintenance of Human Osteoblasts 

Human osteoblast-like cells (MG63) were cultured in conventional tissue culture flasks (250 mL, Greiner, Frickenhausen, Germany) in a humidified atmosphere at 37 °C and 5% CO_2_. Osteosarcoma-derived MG63 cells exhibit features in common with human osteoblast precursors or poorly differentiated osteoblasts. Specifically, these cells produce type I collagen with no or low basal osteocalcin (OC) and alkaline phosphatase (ALP). However, when MG63s are treated with 1,25D, OC expression increases [21,22] and, when the same cells are co-treated with 1,25D and selected growth factors, e.g., LPA/LPA analogues, the levels of ALP markedly increase [8,23,24], a feature of the mature osteoblast phenotype. Consequently, the application of these cells to assess the potential pro-maturation effects of novel biomaterial coatings is entirely appropriate. Cells were grown to confluence in Dulbecco’s modified Eagle’s medium (DMEM)/F12 nutrient mix (Gibco, Paisley, Scotland) supplemented with sodium pyruvate (1 mM final concentration), L-glutamine (4 mM), streptomycin (100 ng/mL), penicillin (0.1 units/mL) and 10% v/v foetal calf serum (Gibco, Paisley, Scotland). The growth media (500 mL final volume) was also supplemented with 5 mL of a 100x stock of nonessential amino acids. Once confluent, MG63s were deemed ready for splitting and seeding for the different experiments detailed in this study. 

### 2.3. Cell Seeding and Treatment at Control and Functionalised Titanium (Ti) 

Unless stated otherwise, Ti discs were seeded with 1ml of a 15 × 10^4^ cells/mL suspension (as assessed by haemocytometry) in serum-free, phenol red-free DMEM/F12 medium (SFCM) supplemented with 100 nM 1,25D. Cells were left for 3 days under conventional cell culturing conditions, and the extent of cell growth and maturation was determined using reliable and robust biochemical assays. 

### 2.4. Functionalisation of Titanium (Ti) Using PDA and FHBP 

The steps taken to achieving an optimal PDA-FHBP coating were initially informed using an inexpensive tissue culture plastic (TCP) model. The final surface finish capable of supporting the greatest extent of osteoblast maturation was deemed optimal. Ti discs were individually placed within wells of 24-well plates. Using 10mM Tris (pH 8.8) as the diluent, FHBP was prepared at a final concentration of 2 μM, and a portion was immediately dispensed onto samples (1 mL/Ti disc). The FHBP concentration was informed from our previous work with this compound for Ti and hydroxyapatite (HA) functionalisations [7,25]. The remaining solution was subsequently combined to solid dopamine hydrochloride (DHC) to achieve a final mass concentration of 2 mg/mL, and the resultant solution was dispensed (1 mL/Ti disc) within a minute of gentle mixing. Solutions were left at an ambient temperature for up to 2 h. At the desired time points, the solution was aspirated, and the Ti samples were rinsed with 2 changes (1ml each time) of cell culture-grade water followed by a final rinse (1 mL/well) with serum-free Dulbecco’s modified Eagle’s medium (DMEM)/F12 nutrient mix (Gibco, Paisley, Scotland). These rinsed Ti samples were then ready for human osteoblast seeding. 

### 2.5. Biochemical Detection of PDA at Ti Surfaces 

Thin films of PDA can be reliably detected using a simple biochemical test using bicinchoninic acid (BCA), as we have reported recently [26]. Briefly, once coated, the Ti specimens were thoroughly rinsed in distilled water, the residual water was removed and then samples were individually placed within wells of 24-well plates. Sample discs were then immersed in 0.25 mL of cell culture-grade water followed by 0.125 mL of the BCA reagent (as prepared according to the manufacturer’s instructions). After a 90 min ambient incubation, sample aliquots (0.1 mL) were subsequently transferred to a 96-well plate, and readings were taken at 540 nm using a multi-well plate reader (FLUOstar OPTIMA, BMG Labtech, Aylesbury, UK) operated from a PC using MARS software. A series of DHC concentrations (0–25 μg/mL) in distilled water enabled extrapolation of the PDA coating, as DHC equivalents, for the different treated surfaces.

### 2.6. An Assessment of PDA-FHBP Coating Stability to a Simulated in vivo Environment 

A supplementary study was conducted to ascertain if the PDA-FHBP coating was retained at the Ti surface following immersion in serum-free DMEM/F12 for a week under conventional cell culturing conditions. To this end, the recovered conditioned media was spiked with 1,25D to a final concentration of 100nM and applied to established lawns of osteoblasts. The remaining discs were also seeded with osteoblasts in the presence of 100nM 1,25D. Following a three-day culture, both models were processed to determine the extent of cellular growth and differentiation.

### 2.7. Surface Wettability Measurements 

Static contact angle measurements were taken using the Model 68-76 Pocket Goniometer PGX+ and PGX+ software (Testing Machines Inc., New Castle, Delaware, USA) operated from a laptop computer. Droplets (4 μL) of cell culture-grade water were used to determine the surface wettability of control and functionalised Ti6Al4V sample discs. For each experiment, 4 contact angle measurements were taken for each of three control and functionalised specimens. A total of three experiments were performed, each on different days. A minimum of 36 measurements were therefore obtained for each of the different Ti surfaces.

### 2.8. Raman Spectroscopy 

To detect the presence of the PDA coating, PDA-coated Ti discs, as well as blank Ti controls, were subject to Raman spectroscopy. Raman spectra were taken from randomly selected points using a Horiba Labram HR Evolution Raman confocal microscope (Horiba Mira Ltd. Basildon, UK) system using an Olympus MPlan N 100X 0.90 NA objective. The Raman signal was excited using a green laser beam (532 nm) at 1% power, corresponding to approximately 0.5 milliwatts, for a 10 s accumulation time. Raman spectra were collected in the region of 500 to 4000cm^−1^.

### 2.9. X-ray Photoelectron Spectroscopy (XPS)

To analyse elements on the surface of Ti and functionalised samples, XPS spectra were taken by using Phi 5600 XPS (Perkin Elmer, Waltham, Massachusetts, MA, USA). Monochromatic X-ray source (Al Kα) beam was used for the data collection. The neutraliser was additionally applied with a ~1-μA current to avoid the surface charging effect on the samples. The deconvolution of high-resolution N1s peaks and the calculation of the atomic percentages were done by using MATLAB-based software (MultiPak V9) developed by Physical Electronics.

### 2.10. Preparation of Osteoblast Monolayers in Multi-well Plates for FHBP-PDA Stability Studies 

Each well of a 24-well culture plate was seeded with 1ml of a 2.5 × 10^4^ cells/mL suspension (as assessed by haemocytometry) in serum-supplemented (10% v/v) DMEM/F12 medium, and the cells were left for 3 days under conventional culturing conditions. The media was subsequently removed and replaced with SFCM, and the cultures were left overnight. The following day, the plates were deemed ready for conditioned media treatment recovered from functionalised Ti surfaces. 

### 2.11. Metabolic Profile of MG63 Osteoblasts in Receipt of FHBP and 1,25D

MG63 osteoblasts were plated at a density of 20,000 cells per well on a Seahorse XFe24 tissue culture plate. FHBP and 1,25D treatments were performed as described above. Cells were analysed using Seahorse XFe Metabolic Cell Energy Phenotype test kits on a Seahorse XFe24 metabolic analyser (Agilent Technologies, Stockport, UK), according to the manufacturer’s instructions. Agilent Wave software (version 2.6 0. 31) was used to analyse data from the metabolic analyses, in order to determine basal and reserve glycolytic and oxidative metabolic rates. Each biological replicate was the average of 5 technical replicates of the same experiment. Results were normalised to the cell numbers, as detailed in Section 2.12 below.

### 2.12. Assessment of Osteoblast Growth at PDA-FHBP-Functionalised Surfaces 

An assessment of the cell numbers was performed using a combination of the tetrazolium compound 3-(4,5-dimethylthiazol-2-yl)-5-(3-carboxymethoxy-phenyl)-2-(4-sulfophenyl)-2H-tetrazolium, innersalt (MTS, Promega, UK) and the electron-coupling reagent phenazine methosulphate (PMS). Each compound was prepared separately in prewarmed (37 °C) phenol red-free DMEM/F12, allowed to dissolve and then combined so that 1 mL of a 1-mg/mL solution of PMS was combined to 19 mL of a 2-mg/mL solution of MTS. A stock suspension of MG63s (1 × 10^6^ cells/mL) was serially diluted in growth medium to give a series of known cell densities down to 25 × 10^3^ cells/mL. Each sample (0.5 mL in a microcentrifuge tube) was spiked with 0.1 mL of the MTS/PMS reagent mixture and left for 45 min within a tissue culture cabinet. Once incubated, the samples were centrifuged at 900 rpm to pellet the cells, and 0.1 mL of the supernatants was dispensed onto a 96-well microtitre plate, and the absorbances were read at 492 nm using a multiplate reader. Plotting the absorbances against the known cell numbers, as assessed initially using haemocytometry, enabled extrapolation of cell numbers for the experiments described herein.

### 2.13. Evaluation of Osteoblast Maturation at PDA-FHBP-Functionalised Surfaces 

An assessment of ALP activity is reliably measured by the generation of p-nitrophenol (p-NP) from p-nitrophenylphosphate (p-NPP) under alkaline conditions. The treatment of cells to quantify ALP activity was similar to that described by us recently [24]. Briefly, the MTS/PMS reagent was removed, and the monolayers were incubated for a further 5 min in fresh phenol red-free DMEM/F12 to remove the residual formazan. Following this incubation period, the medium was removed, and the monolayers were lysed with 0.1 mL of 25 mM sodium carbonate (pH 10.3), 0.1% (v/v) Triton X-100. After 2 min, each well was treated with 0.2 mL of 15 mM p-NPP (di-Tris salt, Sigma, Poole, UK) in 250 mM sodium carbonate (pH 10.3), 1 mM MgCl_2_. Lysates were then left under conventional cell-culturing conditions for 1 h. After the incubation period, 0.1-mL aliquots were transferred to 96-well microtitre plates, and the absorbance was read at 405 nm. An ascending series of p-NP (50–500 μM) prepared in the incubation buffer enabled quantification of product formation. Unless stated otherwise, total ALP activity is expressed as the mean micromolar concentration of p-NP per 100k cells, as extrapolated from the MTS/PMS assay described above.

### 2.14. Statistical Analysis

Unless stated otherwise, results are expressed as the mean plus the standard deviation (SD). Data were subjected to a one-way analysis of variance (ANOVA) to test for statistical significance. In some instances, unpaired *t*-tests (2-tailed) were used to compare means of functionalised and control surfaces. P-values of <0.05 were considered statistically significant. 

## 3. Results 

### 3.1. Short-Term Development of PDA Films are Compatible with Osteoblast Viability 

Using 24-well tissue culture plastic (TCP) plates, we produced PDA coatings over 24, 48 and 72 h under ambient conditions. MG63 cells were seeded into the different PDA-modified wells, cultured for three days and an assessment of cell growth determined at the end of the incubation period. We consistently found modest, yet statistically significant, reductions in cell numbers for each of the PDA-functionalised TCP wells compared to untreated controls (Figure 1A). The reduction in cell number was approximately 11% (*p* = 0.01) for PDA films formed over 24 h. A further reduction at around 16% (*p* < 0.001) occurred for cells exposed to PDA films generated over 48 and 72 h. Extended DHC exposure times generate thick PDA films with greater roughness [27], a feature that can have a negative effect on cell attachment and growth [28]. We subsequently found that TCP wells treated for two hours with DHC resulted in the development of PDA films that were completely compatible with MG63 viability (Figure 1B). We therefore chose a two-hour DHC steeping time in the development of PDA thin films for the remainder of the study. Before moving to Ti, we next examined if a one-pot approach could be taken to functionalise TCP with an FHBP-PDA coating and if this surface modification could support a good osteoblast maturation response to 1,25D. Both FHBP-TCP and FHBP-PDA-TCP supported MG63 maturation, with the latter surface yielding the most superior response (*p* < 0.001) compared to each of the other TCP surfaces (Figure 2). Indeed, the FHBP-PDA-TCP was approximately 2.5 times more effective than FHBP-TCP.

### 3.2. PDA Thin Films are Reliably Detected at Ti Surfaces Using a BCA Assay Reagent 

As reported by us, recently, we were readily able to corroborate PDA film deposition at Ti using the BCA assay reagent [26]. We consistently found that the extent of PDA coverage following a 2-h exposure to DHC was similar for Ti discs exposed to DHC alone or in the presence of 2μM FHBP (Table 1). 

### 3.3. Physicochemical Evidence of PDA Deposition at Ti 

Significant peaks were present at all of the randomly selected points in the Raman spectra at Stokes shifts around the 1410cm^−1^ and 1590cm^−1^. This is in keeping with previous reports (e.g., [29]) for PDA film deposition on Ti (Figure 3A). These peaks were totally absent for the control Ti blank sample. Contact angle measurements taken for PDA-Ti (54.4 ± 3.9°) were significantly less (*p* < 0.0001) than control Ti surfaces (66.7 ± 5.1°), indicating an increase in surface wettability for the former. The contact angle obtained for PDA-FHBP hybrid coated Ti specimens (55.7 ± 5.8°) were similar to PDA-Ti (Table 1). 

The successful PDA coating on the Ti substrate could be confirmed by two distinctive changes in the XPS results. Considering that nitrogen is present in the precursor (i.e., dopamine) of PDA, the increased N1s peak intensity after the coating is the most clear evidence of successful PDA coating on the Ti substrate (Figure 3B). Almost nondetectable N1s (1 at. %) was increased to 5 at. % after the PDA coating (Table 2). The distinctive decrease in the representative element (Ti2p) after the coating is also the evidence of effective PDA deposition. The at. % of Ti2p after the surface modification was decreased from 13% to 4%. 

The presence of FHBP in the PDA-FHBP hybrid coating layer on the Ti substrate could be confirmed by the observation of the noticeable high-resolution P2p peak from the sample (Figure 3B, right top). The calculated at. % of phosphorus in the sample was approximately 1% (Table 2), indicating that the osteoblast maturation effect can be significantly influenced by a trace amount of FHBP. 

### 3.4. Evidence that FHBP Links to the PDA Matrix via a Schiff-Base Reaction

The high-resolution N1s analysis provides insight on the possible linking mechanism between FHBP and PDA, which is assumed to be a Schiff-base reaction (Figure 3C-E). The N1s peaks were deconvoluted with three distinctive amine functional groups by referring to the previous report [29]: primary amine (R-NH_2_) at 401.9 eV, secondary amine (R_2_-NH) at 399.9 eV and tertiary amine (=N-R) at 398.8 eV. As a result, we could identify the primary amine groups (16%) in the PDA structure (Figure 3C), which should be originated from the un-indolised dopamine unit, as proposed by Liebscher et al. [30]. Considering the presence of the ketone group in FHBP, we expected that the Schiff-base reaction can contribute to the tethering of FHBP on the PDA. Regarding this, we confirmed that the PDA-FHBP (hybrid) Ti has a much higher tertiary amine (=N-R) functionality of 14 at. % (Figure 3D) compared to PDA-Ti, which was 8 at. % (Figure 3C). This should be due to the imine bonds (tertiary amine) formation via the Schiff-base reaction (Figure 3E), which strongly supports our assumption.

### 3.5. PDA-FHBP Hybrid Ti Coatings Support a Good Human Osteoblast Maturation Response

Evidence of osteoblast maturation was via total alkaline phosphatase (ALP) activity—specifically, the quantification of p-NP formation from p-NPP. In support of our previous work using FHBP [25], the direct functionalisation of Ti with this LPA analogue generated a surface that was superior to blank Ti at supporting 1,25D-induced MG63 maturation (Figure 4). The increase in p-NP generation for FHBP-Ti compared to blank Ti was significant by approximately 2.2-fold (*p* < 0.0001). When a solution of FHBP was combined to solid DHC and the mixture applied to Ti discs, the resultant PDA-FHBP surface modification was more superior than FHBP-Ti at securing 1,25D-induced osteoblast maturation (Figure 4). Indeed, the increase in p-NP formation was nearly double (*p* < 0.0001) compared to FHBP-Ti. The modification of Ti with a PDA film alone afforded no improvement over blank Ti in securing osteoblast maturation. 

### 3.6. 1,25D and FHBP Drive a Change in the Metabolic Profile of MG63 Cells

Cell differentiation is often accompanied by alterations in the cells’ metabolic profile [31]. In order to assess the effect of 1,25D and FHBP on MG63 cells, we assessed rates and potentials of the cells for glycolytic metabolism and oxidative phosphorylation using a Seahorse XFe24 metabolic analyser (Figure 5). Our results indicate that a three-day treatment with FHBP significantly increases the basal rate of oxidative phosphorylation (Figure 5C) but not the basal rate of glycolysis (Figure 5A). The FHBP/1,25D combination elicited no extra effect on basal oxidative metabolism over the effect of FHBP alone, and 1,25D by itself had no effect on either glycolytic or oxidative metabolism. Interestingly, treatment of the cells with 1,25D, FHBP or a combination elicited a stepwise increase of the total glycolytic capacity of the cells but had no effect on the oxidative capacity. These results indicate that MG63 maturation is accompanied by increased basal mitochondrial oxidative metabolism, which markedly increases the ability of these cells to increase glycolytic rates in response to demand.

### 3.7. PDA-FHBP Hybrid Ti Coatings Exhibit Good Stability to a Simulated in vivo Environment 

In a follow-on experiment, we explored the retention of FHBP to Ti following immersion of FHBP-Ti and PDA-FHBP-Ti in serum-free culture media for one week under cell-culturing conditions. On completion of the incubation time, the conditioned medium was recovered from each of the different Ti specimens, spiked with 1,25D and the resultant solutions applied to established lawns of MG63 cells. Cells were left for three days prior to performing a total ALP activity assay. Each of the different discs that had been immersed were subsequently seeded with MG63 cells, in the presence of 100nM 1,25D, and, also, left to culture for three days prior to performing the ALP assay. The overall outcome from these experiments indicated good stability of the FHBP coating, whether applied alone or in combination with a PDA film. As expected, the co-treatment of established lawns of MG63 cells with 1,25D and FHBP (positive control) stimulated a robust maturation response (Figure 6A) with the level of p-NP from p-NPP at 243 ± 9 μM compared to 99 ± 7 μM for cells treated with FHBP alone and 25 ± 5 μM for MG63s exposed to 1,25D alone. When MG63 cells were treated with 1,25D-spiked conditioned media from each of the different FHBP-functionalised Ti samples, the level of p-NP generated was approximately 50 ± 12 μM, a modest increase in MG63 ALP activity above 1,25D alone but noticeably and significantly less (*p* < 0.0001) than the maturation response incurred to FHBP alone. These data suggest that any FHBP losses from the Ti surfaces were of no significant concern. This was supported for the maturation of MG63 cells to recovered, functionalised Ti discs (Figure 6B). Indeed, the extent of MG63 maturation at both FHBP-Ti and FHBP-PDA-Ti following immersion in medium for one week were very similar to the maturation responses of cells to the same surface treatments without prior culture medium exposure (compare to Figure 4).

## 4. Discussion

Enhancing the biological performance of Ti implants could be a solution to the continuing problem of aseptic TJR loosening. Finding suitable agents as Ti coatings could be one route, particularly if the process of surface functionalisation were achievable in a facile, one-step dip-coating process. An attractive feature of dip-coating resides in the ability of a thin film of agent(s) to be deposited upon a surface regardless of topography or shape [32]. In realising this, we took inspiration from research emerging from Phillip Messersmith’s group, namely the application of PDA-capturing platforms for (bio)material modification by dip-coating materials in mildly alkaline solutions of DHC [13]. Deposited PDA films are amenable to further functionalisation with a wide variety of agents, including peptides, proteins, oligonucleotides and noble metals [12]. The independent reports of PDA-lipid interactions [18,19,20] was of particular interest to us because of our work pertaining to LPA/LPA analogues in the context of calcitriol-induced human osteoblast maturation. Co-treating these cells with LPA and 1,25D results in a synergistic increase in ALP [23], a marker of osteoblast maturation and an enzyme inextricably linked to bone matrix calcification [33]. We have subsequently found that the phosphatase-resistant LPA analogue, FHBP, is much more potent in eliciting a maturation response in human (MG63) osteoblasts [8] and, therefore, a potentially suitable candidate for biomaterial functionalisation. When MG63 cells are co-stimulated with 1,25D and FHBP, we consistently find a clear and synergistic enhancement of alkaline phosphatase (ALP) expression. Given that the cells are producing far greater quantities of ALP in response to these stimuli, the metabolic data presented in this report fit with the energetic demands made by these cells as they make their transition to a more mature phenotype. 

Herein, we report that a PDA-FHBP hybrid film at Ti was capable of supporting a robust and synergistic 1,25D-induced maturation response of MG63 cells. Importantly, this hybrid coating was superior to FHBP-Ti at promoting osteoblast maturation. As a phosphonic acid, FHBP would be predicted to bond robustly to Ti [34,35]. When we steeped Ti in a 2-μM solution of FHBP, the resultant surface supported 1,25D-induced osteoblast maturation akin to what we have observed previously for hydroxyapatite [25]. However, the extent of cellular maturation was further enhanced when FHBP was applied to Ti in a mildly alkaline solution of DHC. Reconstituting DHC in alkaline solution results in the spontaneous polymerisation to PDA, which readily deposits at Ti [13,17]. Within a matter of minutes, the steeping solution began to darken and, by two hours, had adopted a dark brown hue. We were able to corroborate PDA deposition at Ti biochemically using BCA reagent, as reported by us previously [26]. The inclusion of FHBP in the steeping solution did not influence the amount of PDA deposited at the metal surface. The application of Raman spectroscopy provided further evidence for the expected PDA film deposition. Raman shifts at 1410 cm^−1^ and 1590cm^−1^ wavenumber regions are likely attributed to PDA catechol stretching and deformity, respectively, and are in keeping with the polymers’ presence [36]. In addition, the relatively increased N1s peaks from the XPS survey for the PDA-coated Ti provides clear evidence for the successful deposition of PDA. Alteration of Ti surface wettability provided another means of successful PDA deposition; as anticipated from previously published works [37,38,39], the decrease in the contact angle (54.4 ± 3.9°) compared to untreated surfaces (66.7 ± 5.1°) was further evidence for PDA functionalisation of Ti in producing a less hydrophobic surface finish. 

Having established that a one-pot approach to Ti modification with PDA-FHBP resulted in a surface finish that was superior to FHBP-Ti in supporting MG63 maturation, we next examined whether our findings might be attributed to FHBP interacting with dopamine/PDA and/or with 1,25D. Suffice it to say, co-treating MG63 cells with DHC (10 μg/mL) and either 1,25D, FHBP or their combination did not elicit a maturation response (data not shown). 

In taking steps towards the development of novel biomaterial coatings, it is important to consider the stability of the modified surface. One of the first things to consider, for example, is whether the coating can be retained after immersion in a biologically relevant medium. We therefore exposed the different Ti substrates to cell culture media for a week at 37 °C. On completion of the experiment, the bathing solution was removed from each disc, spiked with 1,25D and applied to established lawns of MG63 cells to determine if any of the recovered media could support cellular maturation. The findings gleaned from these experiments indicated a good attachment of the PDA-FHBP coating to Ti. To confirm that the FHBP was still bound to the conditioned discs, MG63s were seeded onto these samples in the presence of 1,25D. On examination of the maturation responses for cells at these recovered samples, the findings were comparable to functionalised Ti that had not been immersed in media for a week. Collectively, it would seem that very little leaching of FHBP had occurred during the incubation period and that the linkage(s) between the coating and the Ti surface were robust.

Whilst there are several studies reporting on the interaction of different lipid species with PDA [18,19,20], we are unclear as to what binding mechanism(s) might exist between FHBP and components of the PDA matrix. In the study by Ding et al. [20], it was found that the polymerisation of dopamine at lipid nanotubes was enhanced where oleic acid was in greater abundance. FHBP bears some structural similarity to oleic acid and would adopt the same negative charge in our experiments as reported by Ding et al. [20] for oleic acid. Of additional note is the finding that oleic acid will combine with dopamine via an amidation reaction [40]. This reaction has been exploited in the fabrication, for example, of PDA-carbon-stabilised sodium ion batteries [41]. Considering that the unindolised dopamine unit, bearing a free amino group, exists within the PDA structure [30], a Schiff-base reaction between dopamine and FHBP (ketone group) may also contribute to the FHBP tethering mechanism, as we proposed in Section 3.4. Whilst we have no data to the contrary, it is conceivable that the incorporation/association of FHBP with PDA is heterogeneous and could include noncovalent supramolecular interactions. For example, entrapped and electrostatic, with multi-site interactions between PDA indole and pendant amines with doubly anionic FHBP. In developing a carbon sphere electrocatalyst, Zhu and colleagues [42] took advantage of a phosphonic acid-dopamine interaction. An intimate link between the two was afforded by an acid-base reaction between phosphoric acid sites and the amine functional group of dopamine. Given that FHBP is classified as a phosphonic acid, it is tempting to speculate that this bioactive lipid might also connect with dopamine, as reported by Zhu et al. [42]. It is also possible that some catechol complexation or H-bonding with the phosphonate could be occurring to help retain the FHBP, and this will be explored in future investigations.

PDA-capturing platforms are gaining significant interest within translational orthopaedics [43]. There is a tangible realisation that PDA could be a dynamic component in the arsenal towards improving the biological performance and, ultimately, the longevity of implantable orthopaedic devices. Using PDA to capture LPA species at Ti could be an important first step in delivering next-generation bone biomaterial design. 

## 5. Conclusions

Phosphonic acids have a strong affinity for metal oxides, including TiO_2_ [34,35], the natural finish of Ti. As anticipated, we were able to functionalise Ti with FHBP, a potent phosphonic acid analogue of LPA. However, when FHBP was applied to Ti in association with a PDA thin film, the functionalised metal was superior to FHBP-Ti in supporting 1,25D-induced osteoblast maturation. The approach we have taken to biologically functionalise Ti is very simple; a facile, dip-coating approach for (bio)material modifications is particularly attractive, since there is no requirement for specialist equipment and the coating process is not constrained to a specific feature [44,45,46]. These are important considerations in realising the fabrication of new implantable Ti technologies, which could now extend to hybrid PDA-bioactive LPA analogue coatings.

## Figures and Tables

**Figure 1 molecules-25-01583-f001:**
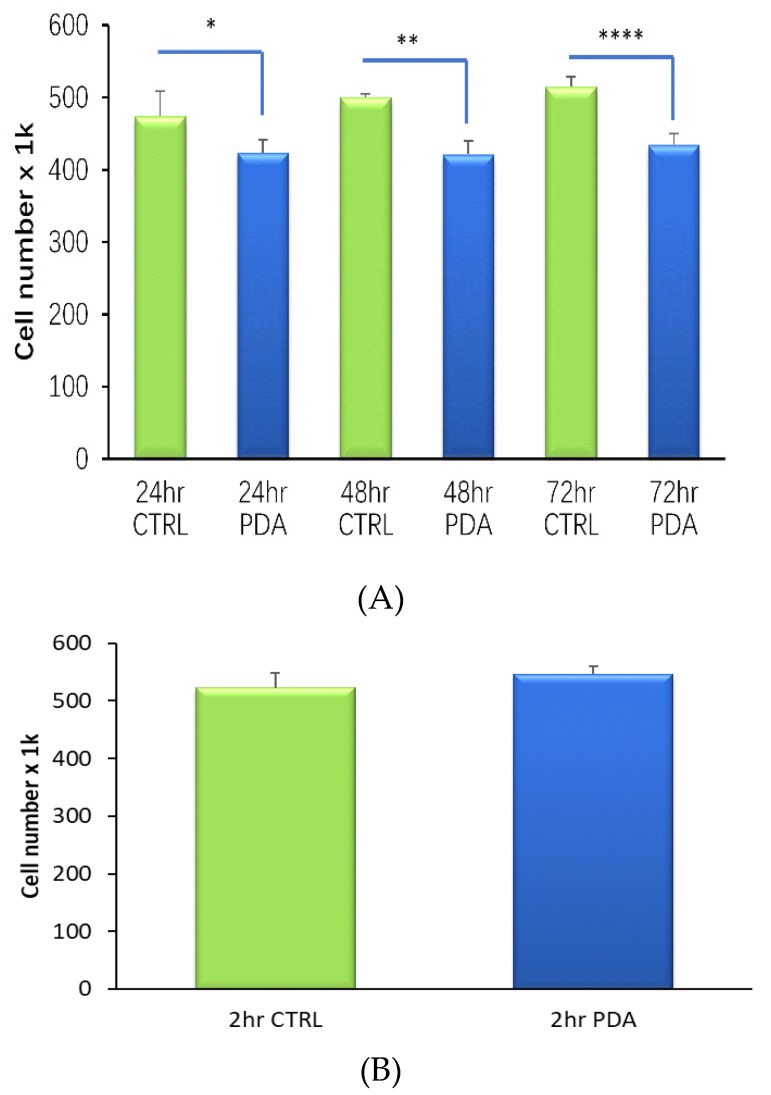
PDA thin films and osteoblast viability. (**A)** Using 24-well tissue culture plastic (TCP) plates, films of PDA were generated over 24, 48 or 72 h. MG63 cells were seeded onto these surfaces and an assessment of cell viability determined after a 3-day culture. Each of the PDA-TCP wells were associated with a modest yet significantly reduced cell viability compared to blank (CTRL) wells (* *p* = 0.01, ** *p* < 0.001, **** *p* < 0.001). (**B)** In contrast, when PDA films were formed over 2 h, the resultant surface finish was compatible.

**Figure 2 molecules-25-01583-f002:**
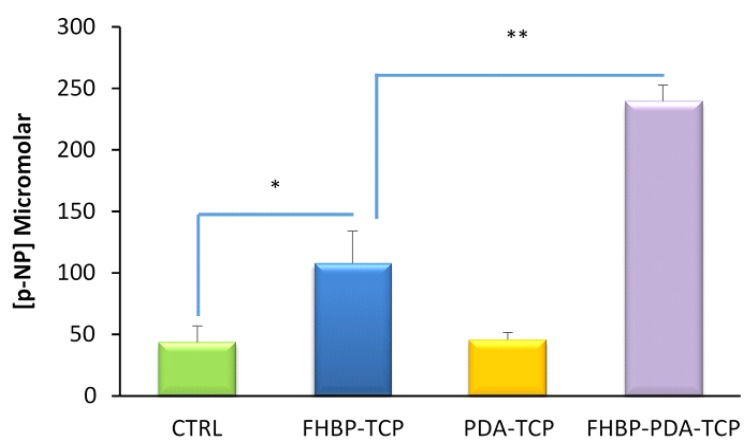
A hybrid FHBP-PDA coating at TCP supports human (MG63) osteoblast maturation. Multi-well (24-well) TCP plates were exposed to either FHBP (2μM), DHC (2mg/mL) or their combination using 10mM Tris (pH 8.5) as the solvent. After 2 h, the plates were aspirated, rinsed and seeded with MG63s in the presence of 1,25D (100nM). After a 3-day culture, an assessment of cellular maturation via total alkaline phosphatase (ALP) assay was performed using p-NPP as the substrate and quantification of p-NP generation. Treating TCP with FHBP resulted in a surface that was significantly better at promoting MG63 maturation compared to blank (CTRL) wells (* *p* < 0.001). Hybrid surfaces were even better than FHBP-TCP in supporting 1,25D-induced maturation (** *p* < 0.0001). The data depicted are pooled from three independent experiments and are expressed as the mean micromolar concentration of p-NP per 100k cells plus the standard deviation. For each bar, N = 18.

**Figure 3 molecules-25-01583-f003:**
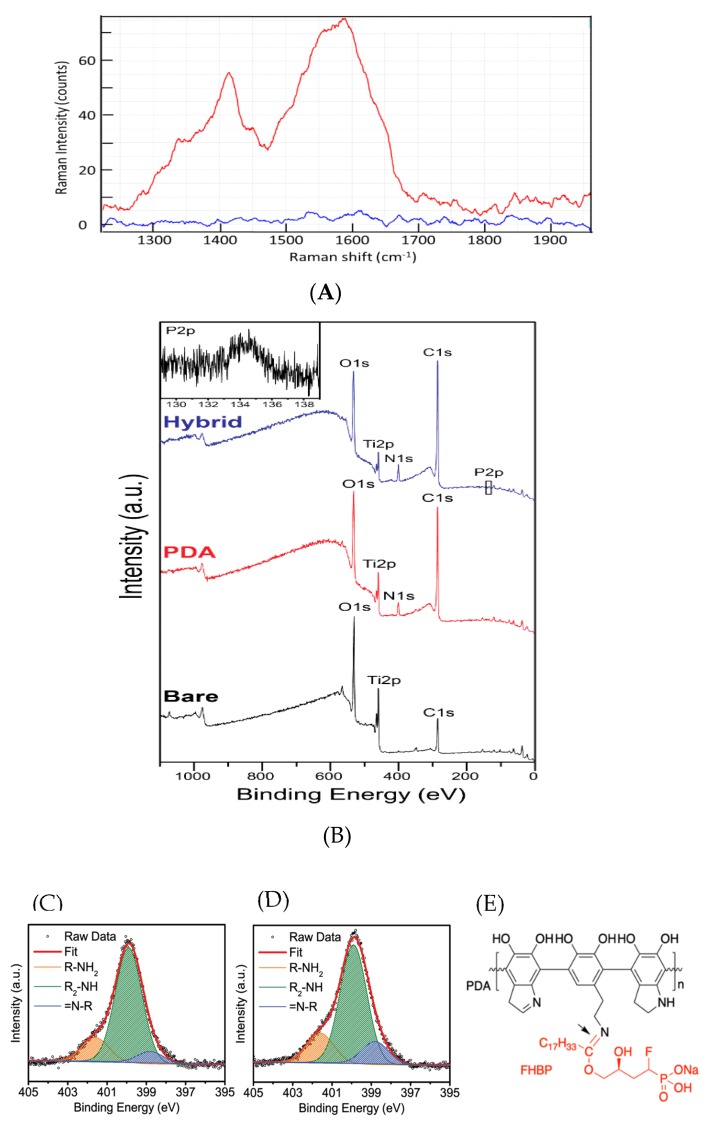
Physiochemical evidence of PDA-FHBP deposition at Ti. (**A**) Typical Raman spectra of a polydopamine-coated titanium disc sample (upper trace) and a blank control titanium disc (lower trace) after removal of the background signal. The upper trace is displaced vertically by +10 units for clarity of comparison. (**B)** For X-ray photoelectron spectroscopy (XPS) of PDA-Ti, sample discs were exposed to DHC (2 mg/mL), FHBP (2 μM) or their combination for 2 h using 10 mM Tris (pH 8.5) as the solvent. The XPS data supports the successful deposition of PDA at Ti, as the N1s signal is only evident for the functionalised surfaces. In addition, the Ti2p signal is reduced following DHC/DHC-FHBP exposure. The corresponding high-resolution peak corroborating FHBP integration (P2p) is also presented for the hybrid surface. The data depicted are a representative from three replicate samples. Deconvoluted high-resolution N1s peaks are from PDA-Ti (**C**) and PDA-FHBP (hybrid) Ti (**D**). The data demonstrates the higher atomic percentage of tertiary amine functional groups in PDA-FHBP (hybrid) Ti, which should be due to the Schiff-base reaction between PDA and FHBP. Expected conjugation chemistry between PDA and FHBP via the Schiff-base reaction (**E)**.

**Figure 4 molecules-25-01583-f004:**
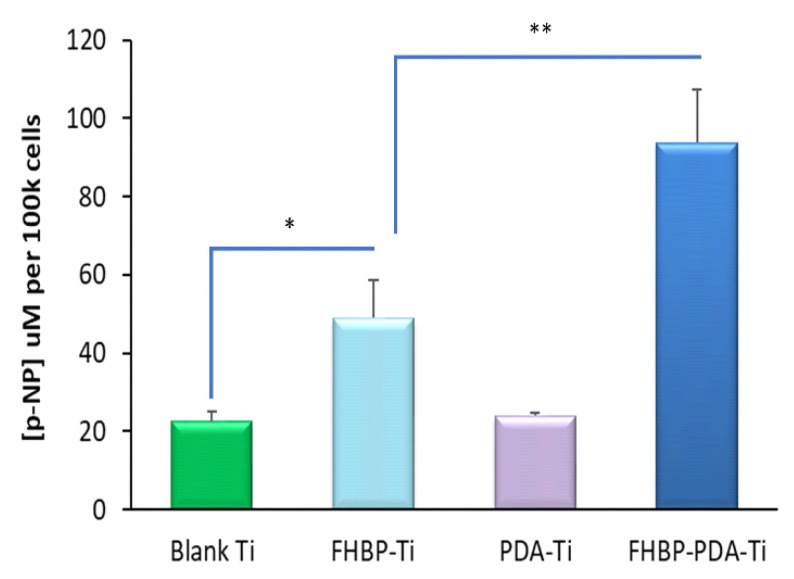
Human osteoblast maturation at FHBP-PDA Ti. Solid Ti discs were exposed to DHC (2 mg/mL), FHBP (2 μM) or their combination for 2 h using 10mM Tris (pH 8.5) as the solvent. After 2 h, the plates were aspirated, rinsed and seeded with MG63s in the presence of 1,25D (100 nM). After a 3-day culture, an assessment of cellular maturation via total alkaline phosphatase (ALP) assay was performed using p-NPP as the substrate and quantification of p-NP generation. Treating Ti with FHBP resulted in a surface that was significantly better at promoting MG63 maturation compared to blank Ti (* *p* < 0.001). Hybrid surfaces were even better than FHBP-Ti in supporting 1,25D-induced maturation (** *p* < 0.0001). The data depicted are pooled from three independent experiments and are expressed as the mean micromolar concentration of p-NP per 100k cells plus the standard deviation. For each bar, N = 18.

**Figure 5 molecules-25-01583-f005:**
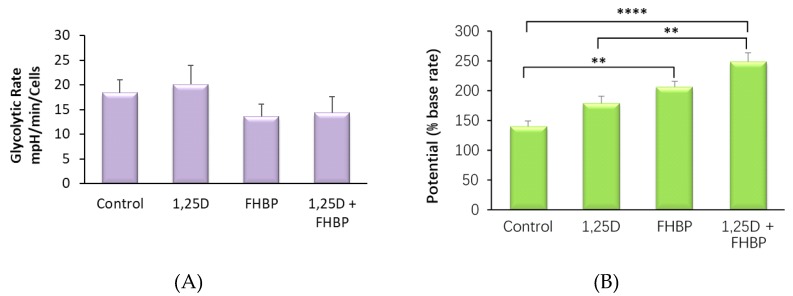
MG63 metabolic phenotype changes during differentiation. MG63 cells were treated with 1,25D, FHBP or a combination in an identical way to other experiments and assayed with a Seahorse XFe24 metabolic analyser, using the Cell Energy Phenotype test kit. (**A**) Baseline glycolytic rate, derived from basal extracellular acidification rate (ECAR). (**B**) Glycolytic potential derived from oligomycin-induced maximal ECAR expressed as a percentage of the base rate. (**C**) Base rate of mitochondrial metabolism derived from basal oxygen consumption rate (OCR). (**D**) Mitochondrial potential derived from FCCP-induced maximal OCR expressed as a percentage of basal OCR. All results were normalised for cell count using an MTS-PMS assay. Results are expressed as the mean plus SEM (n = 4). * *p* < 0.05, ** *p* < 0.01, **** *p* < 0.0001 (one-way ANOVA with Tukey’s post hoc test).

**Figure 6 molecules-25-01583-f006:**
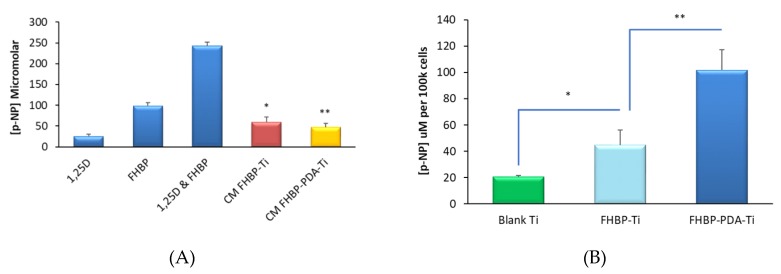
FHBP is retained at Ti following exposure to a simulated in vivo milieu. FHBP-Ti and FHBP-PDA-Ti discs were immersed in culture medium for a week in a humidified atmosphere at 37 °C. Once incubated, the conditioned media (CM) was removed, spiked with 1,25D (100 nM) and applied to established monolayers of MG63 cells. Each of the recovered discs were seeded with MG63s in the presence of 1,25D. Both culture systems were left for 3 days prior to an assessment of total ALP activity to ascertain the extent of cellular maturation. As anticipated, the co-treatment of cells with FHBP and 1,25D led to a clear increase in ALP activity, as indicated by the greater concentration of p-NP (**A**). However, the CM for both the functionalised Ti samples did not support such a robust maturation response, with the p-NP concentration being significantly less than for FHBP alone (* *p* < 0.001). Conditioned discs retained their ability to support a good maturation response, with clear increases in p-NP generation for FHBP-Ti (* *p* < 0.0001) and FHBP-PDA-Ti (** *p* < 0.0001) compared to blanks (**B**).

**Table 1 molecules-25-01583-t001:** Ti contact angle measurements and PDA detection. When solid Ti discs were exposed to DHC (2 mg/mL), FHBP (2 μM) or a combination of these agents for 2 h, the resultant PDA-Ti and FHBP-PDA-Ti exhibited a significant reduction in the contact angle (* *p* < 0.001) compared to blank and FHBP-Ti specimens. Treating PDA-Ti and the hybrid-coated Ti with BCA reagent afforded biochemical detection of PDA deposition, expressed as μg/mL DHC equivalents. The inclusion of FHBP had no influence upon PDA accumulation at Ti.

Surface Modification	Blank Ti	FHBP-Ti	PDA-Ti	FHBP-PDA-Ti
Contact angle	66.7 ± 5.1°	65.3 ± 2.8°	54.4 ± 3.9° *	55.7 ± 5.8° *
PDA coating-μg/mL DHC equivalents			2.1 ± 1.04	2.37 ± 0.87

**Table 2 molecules-25-01583-t002:** Elemental analysis of control and functionalised Ti. The atomic concentrations of elements for bare Ti, PDA-Ti and FHBP-PDA (hybrid) Ti confirm the successful deposition of PDA upon Ti. Given that nitrogen is present in the dopamine precursor, the clear increase in N1s and corresponding decrease in Ti2p support PDA functionalisation. The trace amount of phosphorous (P2p) for the hybrid surface supports FHBP-PDA modification of Ti.

Sample	C1s (at. %)	N1s (at. %)	O1s (at. %)	P2p (at. %)	Ti2p (at. %)
Bare	50	1	36	–	13
PDA	63	5	28	–	4
Hybrid	67	6	23	1	3

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
