# Peer review of "Polydopamine-Lysophosphatidate-Functionalised Titanium: A Novel Hybrid Surface Finish for Bone Regenerative Applications"

_molecules, 2020, doi:10.3390/molecules25071583_

Round 1

Reviewer 1 Report

In this work submitted by Mansell et al. reports a facile, dip coating approach for modification of Ti implants by using dopamine and (3S)1-fluoro-3-hydroxy-4-(oleoyloxy)butyl-1-phosphonate (FHBP), a phosphatase-resistant analogue of lysophosphatidic acid. The engineered Ti can cooperate with calcitriol to promote human osteoblast formation and maturation. This work is organized well and the results largely support their conclusion. I would recommend its publication after minor revision with the following concerns addressed.

  • A longer time of incubation in dopamine solution would typically produce thicker PDA coating. Why the thickness would affect the number of cells (Figure 1)? And what is the thickness of FHBP-PDA on TCP?
  • In view of materials, can the authors confirm that FHBP was loaded into the coating matrix? The signal in Figure 3 seems very weak, and cannot convince me too much. In addition, what is the loading percentage of FHBP present in the coating?
  • I was confused by the use of BCA kit in this experiment (Page 7, 3.2 section)? BCA is commonly used for determining total protein. Where is the BCA result then?
  • Relevant references regarding PDA surface modification should be also cited, such as Angew. Chem. Int. Ed. 2018, 57, 1-7; ACS Nano 2016, 10, 12, 11066-11075. These would help the readers to understand the nature of PDA.
  • The authors stated that FHBP was linked to PDA via Shiff-base reaction. But I would recommend that the authors also discuss the presence of non-covalent bonds, such as absorption via supramolecular interactions, unless the authors can provide data to rule out the possibilities of these interactions.

Reviewer 2 Report

The paper by Baldwin and collegues addresses an interesting topic: how to enhance the longevity of titanium implants?

They propose a clever and easy way to coat titanium surfaces with a combination of LPA and FHBP. Such a combination enhances human osteblasts maturation and the coating is still present and able to support osteoblasts' maturation after a one-week incubation in culture medium.

It is not clear if the coating will actually enhance the longevity of titanium implants in a clnical setting. In a previous similar work on hydroxyapatite the same group performed also "stress tests [...] to evaluate coating survivorship after exposure to mechanical and thermal insults that are routinely encountered in the clinical environment." Considering only the stability after 1 week of submersion is not enought. Moreover the authors wrote about testing the possible prolonged shelf life of LPA-FHBP coated titanium, but no specific test was performed. Please add data on this topic or at least discuss it.

The material and methods section and the results are somehow mixed up. Many details on methods are given in the figure legends in the results section. The same for the results themselves: the results should be described in the maintext, not in the legends. Please describe the methods in the material and methods section and the results i the resultsì maintext. Use the figure legens to describe what is presented in the figures, not the results.

I am not sure about what the image at the bottom of table 1 represents, since there is no description neither in the text nor in the caption.

On page 9: "The increase in p-NP generation for FHBP-Ti compared to blank Ti was significant by approximately 3.3-fold (p<0.0001)." But looking at the figure, the fold increase is lower.

Please check the text for typo errors:

  • I have the feeling that some "µ" went lost (i.e. "µM" is written " M")
  • correct "approach" on page 5, line 5 from the bottom
  • correct "representative" on page 8, line 3 from the bottom of the main text
